

# Ocean alkalinity enhancement in an open ocean ecosystem: Biogeochemical responses and carbon storage durability

Allanah Joy Paul[1], Mathias Haunost[1], Silvan Urs Goldenberg[1], Jens Hartmann[2], Nicolás Sánchez[1], Julieta
Schneider[1], Niels Suitner[2], Ulf Riebesell[1]

[1] Biological Oceanography, GEOMAR Helmholtz Centre for Ocean Research Kiel, Düsternbrooker Weg 20, 24105 Kiel, Germany

[2] Institute for Geology, University of Hamburg, Bundesstrasse 55, 20146 Hamburg, Germany

*Correspondence to*: Allanah J. Paul (allanah.paul@gmail.com)



**Abstract.** Ocean alkalinity enhancement (OAE) is considered for the long-term removal of gigatons of carbon dioxide ($CO_2$)
from the atmosphere to achieve our climate goals. Little is known, however, about the ecosystem-level changes in

biogeochemical functioning that may result from the chemical sequestration of $CO_2$ in seawater, and how stable the
sequestration is. We studied these two aspects in natural plankton communities under carbonate-based, $CO_2$-equilibrated OAE
in the nutrient-poor North Atlantic. During a month-long mesocosm experiment, the majority of biogeochemical pools,
including inorganic nutrients, particulate organic carbon and phosphorus as well as biogenic silica, remained unaltered across
all OAE levels of up to a doubling of ambient alkalinity (+2400 µeq kg$^{-1}$). Noticeable exceptions were a minor decrease in

particulate organic nitrogen and an increase in the carbon to nitrogen ratio (C:N) of particulate organic matter in response to
OAE. Thus, in our nitrogen limited system, nitrogen turnover processes appear more susceptible than those of other elements
leading to decreased food quality and increased organic carbon storage. However, alkalinity and chemical $CO_2$ sequestration
were not stable at all levels of OAE. Two weeks after alkalinity addition, we measured a loss of added alkalinity and of the
initially stored $CO_2$ in the mesocosm where alkalinity was highest (+2400 µeq kg$^{-1}$, $\Omega_{aragonite}$ ~10). The loss rate accelerated

over time. Additional tests showed that such secondary precipitation can be initiated by particles acting as precipitation nuclei
and that this process can occur even at lower levels of OAE. In conclusion, on the one hand, our study under carbonate-based
OAE where the carbon is already sequestered, the risk of major and sustained impacts on biogeochemical functioning may be
low in the nutrient-poor ocean. On the other hand, the durability of carbon sequestration using OAE could be constrained by
alkalinity loss in supersaturated waters with precipitation nuclei present. Our study provides evaluation of ecosystem impacts

of an idealised OAE deployment for monitoring, reporting and verification (MRV) in an oligotrophic system. Whether
biogeochemical functioning is resilient to more technically simple and economically more viable approaches that induce
stronger water chemistry perturbations remains to be seen.

*Key words: carbon dioxide removal, subtropical North Atlantic, pelagic ecosystem, negative emission technologies (NETs),*

*plankton community*



## 1 Introduction

There is growing recognition that removal and long-term sequestration of carbon dioxide is needed in order to remain within the carbon budget of cumulative emissions to restrict warming to 1.5°C (Rogelj et al., 2018). While drastic $CO_2$ emissions

reductions are essential and urgent, certain anthropogenic activities have a residual carbon emission that cannot be eliminated via emissions reduction alone. These residual emissions hence require active carbon dioxide removal (CDR) to reach Net Zero $CO_2$ targets.

A variety of ecosystem based and technological approaches have been proposed to remove and durably store carbon (C). Ocean

Alkalinity Enhancement (OAE) is one approach currently under investigation for its potential to chemically bind $CO_2$ in seawater (Hartmann et al., 2013; Kheshgi, 1995; Rau and Caldeira, 1999). Through concurrent shifts in multiple chemical equilibria, OAE increases the amount of dissolved inorganic carbon (DIC) that can be stored in seawater. This essentially mimics natural rock weathering that is the major alkalinity input to the ocean (Mackenzie and Garrels, 1966; Middelburg et al., 2020) and happens on timescales of 10 000 – 100 000 years. Each year this natural chemical weathering process already

sequesters around 0.25 Gt C (Hartmann et al., 2009). Modelling studies have indicated a high potential for acceleration of this process using NETs to sequester carbon on the gigaton scale (Keller et al., 2014) and thereby stabilise future atmospheric $CO_2$ concentrations (Ilyina et al., 2013). Additionally, OAE increases the capacity of seawater to buffer changes in pH, possibly counteracting ocean acidification in vulnerable ecosystems (Feng et al., 2016; Mongin et al., 2021)

OAE can be achieved on much shorter time-scales in many ways using a range of materials and implementation strategies (NASEM, 2022). Some approaches propose utilising widely available materials such as natural minerals or industrial by-products (Renforth and Henderson, 2017). Both carbonate and silicate-based minerals are abundant and commonly considered as alkalinity sources. These materials also contain biologically active elements (micronutrients, trace metals e.g. nickel, magnesium, iron) which leach from the mineral concurrently with alkalinity (Montserrat et al., 2017). Despite higher energy

production costs, electrochemically generated hydroxide solutions are chemically simpler and, as such, do not contain these additional components (NASEM, 2022). OAE can be implemented in a non-equilibrated fashion (Hartmann et al., 2023), where the chemical $CO_2$ uptake occurs at the rate of natural gas exchange between the atmosphere and the ocean. This approach initially induces substantial perturbations to seawater carbonate chemistry (high pH, low $pCO_2$, high calcium carbonate saturation state or $\Omega_{CaCO3}$) that gradually diminish with progressive $CO_2$ in-gassing and water mixing, which can take months

to years (Bach et al., 2023; Jones et al., 2014). In addition, seawater pH and calcium carbonate saturation may reach thresholds where spontaneous precipitation of carbonates, and hence a loss of alkalinity may occur (Chave and Suess, 1970). Ideally, OAE is implemented where the chemical binding of $CO_2$ has already occurred in the dissolved inorganic carbon (DIC) pool before release into the ocean. In other words, the seawater $pCO_2$ is already equilibrated with the atmospheric $pCO_2$ and the carbonate system changes in the implementation area are stable and moderate. This approach has the additional benefit that





the chemically sequestered carbon can be easily quantified, a valuable consideration for monitoring, reporting and verification (MRV) requirements. Consequently, the way how OAE is applied will affect potential biological responses in marine ecosystems. Many economically attractive and technologically feasible OAE approaches generate increased complexity in the concurrent chemical perturbations, which may increase the probability of ecosystem responses and greater uncertainty in the potential environmental impacts. Crucial factors for ecosystem-level responses include the type, magnitude and stability of

seawater chemistry perturbations induced.

       Two decade of research into the impacts of ocean acidification on marine phytoplankton has demonstrated physiological sensitivity to changes in seawater carbonate chemistry, in particular $CO_2$ concentration and pH. This sensitivity often differs between distinct phytoplankton groups (Kroeker et al., 2013; Paul and Bach, 2020), hence the mere perturbations in the seawater $CO_2$ system induced by OAE may cause species-specific responses (Bach et al., 2019a; Ferderer et al., 2023).

Enhanced alkalinity changes the availability of the two key substrates for carbon fixation: bicarbonate ions ($HCO_3^-$) and $CO_2$, both of which can be utilized by phytoplankton. To improve carbon uptake, many phytoplankton groups additionally developed carbon concentrating mechanisms, which differ between species in terms of their efficiency and energetic costs (Colman et al., 2002; Giordano et al., 2005). Changes in seawater $CO_2$ due to OAE may thus influence the phytoplankton composition by turning the carbon uptake conditions into more or less favourable for certain phytoplankton groups or species, respectively

(Pierella Karlusich et al., 2021). Furthermore, OAE increases the seawater pH, and thus reduces potential negative impacts of higher [$H^+$] on plankton physiology, particularly on calcifying phytoplankton such as coccolithophores (Bach et al., 2015; Paul and Bach, 2020). In theory, calcifiers may benefit from this increase in pH as a result of both lower energetic costs for calcification and reduced chemical dissolution of their calcium carbonate structures. While OAE may benefit calcifying organisms, the process of calcification again consumes alkalinity, so that the proliferation of calcifying phytoplankton would

reduce the efficiency of OAE as a CDR method. Overall, OAE may shift the composition of marine plankton communities, with potential consequences for key biogeochemical processes including nutrient cycling and primary production, as well as carbon storage and sequestration in the ocean. Currently, the potential of OAE is assessed on modelling studies that, as previously described, predict a reasonable potential of this method to counter climate change. Field-based assessments to evaluate potential impacts on nature and verify its potential are still in their infancy. With this study we aim to help change

that.

       Here, we carried out a mesocosm experiment to investigate how a nutrient-poor plankton community responds to carbonate-based, $CO_2$-equilibrated OAE. This idealised OAE represents a best-case scenario from both a CDR verification and an environmental impact perspective. We focussed on changes in the biogeochemical element pools that would be induced by shifts in primary producers in a natural plankton assemblage from the subtropical eastern North Atlantic Ocean. We assessed

the risk of potentially undesired changes in nutrient and organic matter partitioning and durability of $CO_2$ storage due to loss





via precipitation. This type of information is essential for supporting evidence-based decisions on a safe deployment of OAE within portfolio of CDR approaches.

## 2 Methods

### 2.1 Mesocosm experiment design and set-up

Nine mesocosms were deployed in the Taliarte Harbour on the eastern coast of Gran Canaria in the subtropical North Atlantic Ocean. The mesocosms were supported by floating frames from which a flexible bag of 4 m length was suspended and enclosed at the bottom with a conical shaped sediment trap (Bach et al., 2019b; Goldenberg et al., 2022, see photos in Fig. S1). These were filled on September 10$^{th}$ 2021 with water pumped from outside the harbour from 2-10 m deep using a peristaltic pump and large diameter hoses to gently transfer the plankton organisms into the mesocosm bags (14 m$^3$ h$^{-1}$, KUNZ SPF60,

Flexodamp FD-50). The collected water was distributed equally across all mesocosms during the filling period using digital flow meters with final volumes ranging between 8001 - 8051 L.

A gradient design was used in this experiment with seawater alkalinity ranging between ambient (0 µeq kg$^{-1}$ added alkalinity, OAE0) and 2400 µeq kg$^{-1}$ additional alkalinity (OAE2400). The alkalinity levels increased in equal intervals of 300 µeq kg$^{-1}$

across nine mesocosms (OAE0, OAE300, OAE600, OAE900, OAE1200, OAE1500, OAE1800, OAE2100, OAE2400). We selected this wide range of alkalinity levels to specifically examine the potential for any adverse impacts, with the upper limit of +2400 µeq kg$^{-1}$ informed by the saturation state of calcite ($\Omega_{Ca}$), which when $\Omega_{Ca}$ = 15-20 can lead to spontaneous precipitation (T = 25°C, Morse and He 1993). This upper limit also corresponded to double the ambient seawater alkalinity in all mesocosms measured on Days 1-3 (2397.7 ± 3.6 µeq kg$^{-1}$, mean ± s.d.). Seawater alkalinity in the mesocosms was enhanced

by addition of 22 kg of NaHCO$_3$ and 22 kg of Na$_2$CO$_3$ solution on Day 4. Each salt was weighed and dissolved separately into 22 kg of deionised water with the precise amounts corresponding to the alkalinity to be added with final treatments. Salinity was adjusted to 35 g kg$^{-1}$ using NaCl, taking into consideration the amount of alkalinity adjustment. Finally, the respective alkalinity solutions were added to the mesocosms. CTD profiles with the attached pH sensor (see Sect. 2.2 for CTD and sensor details) were taken directly after addition to ensure that the alkalinity adjustment was successful, in addition to discrete samples

taken for DIC analyses from the three highest OAE treatments (1800, 2100, 2400 µeq kg$^{-1}$).

### 2.2 Mesocosm sampling and maintenance

The mesocosms were regularly sampled from surrounding platforms that were attached to the pier, with daily sampling from Day 1-3, followed by sampling every 2$^{nd}$ day from Day 3 onward. An overview of experimental activities and the timeline is provided in Fig. 1. Discrete water samples for nutrient, particulate matter, and Chlorophyll *a* analyses were collected between

08:00-10:50 local time. These water samples were collected in multiple deployments depth-integrating sampling tubes between the surface and 2.3 m depth, and then filled into 10 L plastic cannisters for transport to the laboratory and subsampling. CTD





profiles (CTD 60M, Sea and Sun Technology), including additional sensors for dissolved oxygen ($O_2$), pH and photosynthetically active radiation (PAR) were carried out between 0.5 – 3.0 m deep after the discrete water sampling and concluded around 11:00.


Mesocosms were cleaned inside, using a pool brush, and outside by scuba divers, including the sediment trap funnels, to remove wall growth and minimise any shading effects (see Fig. S1 in Supplementary Material for photos) during the study period. An isotope tracer ($^{13}$C-DIC) was added in trace amounts to assess the transfer of carbon throughout the food web.

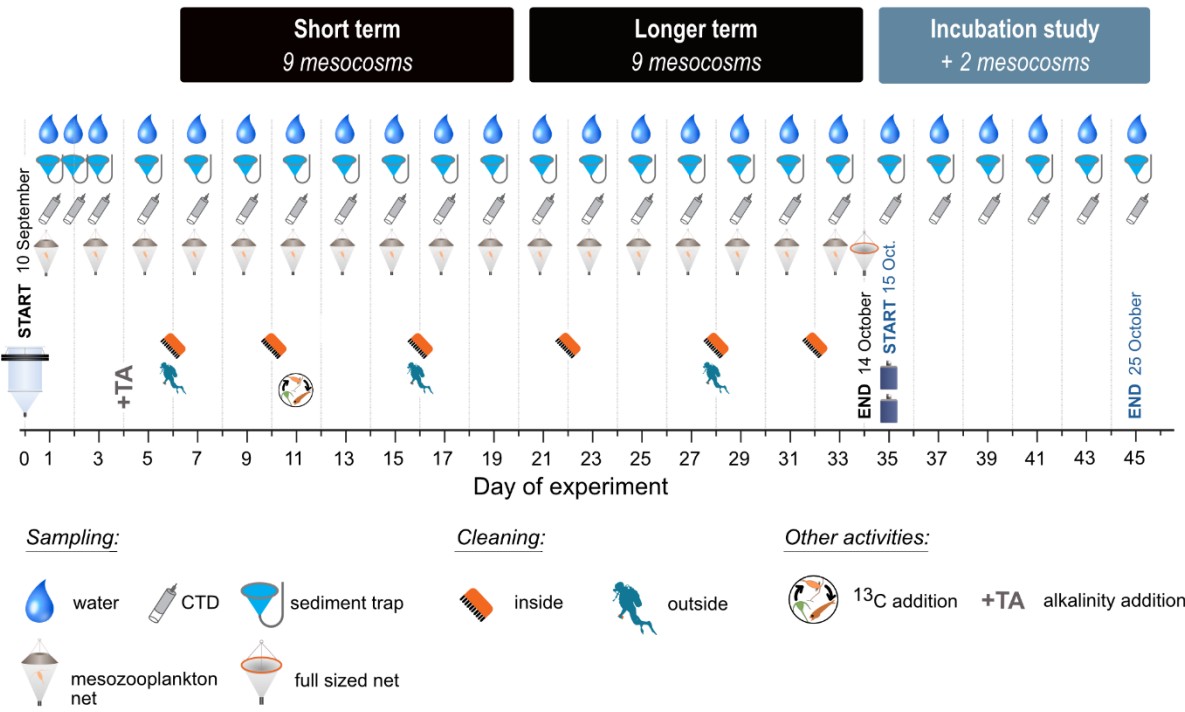

**Fig. 1: Experimental time line for the mesocosm and incubation studies indicating sampling, cleaning and other activities.**

**2.3 Incubation study design and sampling procedures**

We carried out an incubation study starting on Day 35 to investigate potential mechanisms of a key observation that emerged during the mesocosm study from Day 21 onward: In the highest manipulated mesocosm, OAE2400, a decrease in alkalinity was observed and calcium carbonate precipitation occurred in form of white flakes that collected at the mesocosm walls and

in the sediment trap. The alkalinity decreased continuously until the end of the experiment and even fell below concentrations of the second highest mesocosm (OAE2100), in which, however, no alkalinity decrease or carbonate precipitation were observed. The aim of this side-study was to understand why the alkalinity decrease and carbonate precipitation only occurred in these mesocosms and whether this was due to precipitation nuclei, which (i) were already present in the seawater, (ii)



occurred only in that single mesocosm due to a wall effect, or (iii) whether they were actually caused by the high alkalinity

alone.

For the incubation study, the two highest alkalinity levels (OAE2400 and OAE2100) were chosen in order to trigger precipitation in at least one of these treatment levels. For each OAE level, mesocosm water that already had enhanced alkalinity (pre-adjusted) was used, as well as collected fresh oligotrophic seawater at 4 meters depth in the vicinity of the Taliarte Harbour

(27° 59′ 24″ N, 15° 22′ 8″ W; east coast Gran Canaria, Spain, TA ≈ 2411 µeq kg$^{-1}$, S ≈ 36.6, T ≈ 23°C, pH ≈ 8.15, on 30$^{th}$ September 2021). The seawater was filtered through a 55 µm gauze to exclude larger organisms and particles, and the alkalinity was adjusted to +2100 and +2400 µeq kg$^{-1}$ using $NaHCO_3$ and $Na_2CO_3$ as described for the mesocosm study (Sect. 2.1). White crystalline material was collected from the walls of the mesocosm and was added as seeding material (120 mg wet weight to each treatment). To test the effect of precipitation nuclei, both mesocosm and fresh seawater treatments were further divided

into "seeded" and "non-seeded" treatments. Each treatment was carried out in duplicate in 8 L plastic bottles (PET).

All bottles were submerged at about 1.5 m depth in the Taliarte Harbour, attached to the pier in random order. Samples were taken in the morning at the start of the experiment (Day 0) and every 2nd day, with final sampling on Day 10. Sterile filtered (polyethylsulfone cartridge, pore size 0.2 µm, Sarstedt, Germany) sub-samples of 150 mL were taken to examine carbonate

chemistry variables and alkalinity changes during the 10-day study period and the flexible PET bottles were compressed to minimise headspace. Temperature, pH, and salinity, were analyzed with a WTW multimeter (MultiLine® Multi 3630 IDS, pH-probe: SenTix 940 pH-electrode, conductivity: TetraCon 925 cell, Xylem). The pH-probe was calibrated with WTW buffer solutions in 4 steps (1.679 - 9.180 at 25°C) according to NIST/PTB and values were recalculated according to (Badocco et al., 2021). For the TetraCon 925 cell, 0.01 mol L$^{-1}$ KCl calibration standards for conductivity cells (WTW, tracibleto NIST/PTB)

were used. Alkalinity measurements were carried out as for mesocosm samples (Sect. 2.4.2).

### 2.4 Measured variables

### 2.4.1 Nutrient concentrations

Subsamples for nutrients (combined nitrate + nitrite, nitrite, phosphate, silicate) were collected in acid cleaned polycarbonate bottles directly at the pier, and filtered (0.45 µm Sterivex, Merck) before analysis spectrophotometrically using an

Autoanalyser (QuAAtro autoanalyzer, SEAL Analytical) using an autosampler (XY2 autosampler, SEAL Analytical) and a fluorescence detector (FP-202, JASCO) according to Hansen and Koroleff, (1999). Limits of detection (LOD) were determined daily as the variability ($\mu \pm 3\sigma$) in the concentration of the lowest standard in the calibration series because this was deionised water which should have contained no nutrients. Nitrite was very low during the study (< 0.04 µmol L$^{-1}$) hence we report combined [NO$_x$] that was directly quantified spectrophotometrically as the sum of nitrate + nitrite.



### 2.4.2 Carbonate chemistry

Samples for dissolved inorganic carbon (DIC) and total alkalinity (TA) analyses were collected directly from the integrating water samplers in 250 mL glass flasks. Separate subsamples for DIC and TA were filtered (polyethylsulfone cartridge, pore size 0.2 µm, Sarstedt, Germany) before analysis to remove particulate inorganic carbonate. Care was taken during sampling and filtering to keep the samples bubble free and minimise contact with the atmosphere.

TA concentrations were determined by potentiometric titration with HCl using a Metrohm 862 Compact Titrosampler, Aquatrode Plus (Pt1000), and 907 Titrando unit, as described in Chen et al. (2022). Variability in TA measurements over the first three sampling days before alkalinity manipulation was maximum 6.5 µmol kg$^{-1}$ (indicated by the standard deviation of TA on Days 1-3). DIC concentrations were analysed by infrared absorption (LI-COR LI-7000, AIRICA system, MARIANDA, Kiel). Seawater standards were used to determine accuracy of TA and DIC analyses (Batches 143, 190; Dickson, 2010). DIC variability over the first three sampling days for a given mesocosm, indicated by the standard deviation of DIC on Days 1-3, was maximum of 10.2 µmol kg$^{-1}$.

K1 and K2 equilibrium constants from Lueker et al. (2000) were used to calculate other carbonate system variables (e.g. pCO$_2$, CO$_2$, $\Omega_{Ar}$, pH$_{sw}$) as these agree well with direct measurements of DIC and TA and pCO$_2$ (Dickson et al., 2007) with a boron (B$_T$ value) taken from Uppström (1974). The seawater scale is used to report pH as pH$_{sw}$.

### 2.4.3 Particulate matter analyses

Subsamples for total particulate carbon (TPC), particulate organic carbon (POC), particulate organic nitrogen (PON), and particulate organic phosphorus (POP) analyses were taken from the pooled water in the cannisters. Particles were collected by vacuum filtration (p < 0.3 mbar) onto glass fibre filters (nominal pore size 0.7 µm, Whatman GF/F) that had been pre-combusted at 450°C for 6 hours to remove any organic material. Filtered volumes ranged between 500 and 1500 mL. Filters for TPC, POC and PON were dried at 60°C before packing into tin capsules and analysis, with an additional acidification step prior to drying for POC filters to remove inorganic particulate carbon. All samples were analysed on an elemental analyser (Flash EA, Thermo Fisher) connected to a mass spectrometer (Delta V Advantage Isotope Ratio MS, Thermo Fisher) by a Conflo IV interface (Thermo Fisher). Acetanilide and caffeine were used as standard reference materials for the CN content. PIC concentrations are reported as the difference in carbon between the TPC concentration and the POC concentration (acidified filter).

POP concentrations were determined from the filter samples by oxidising the organic phosphorus in the collected particles to phosphate using an oxidising decomposition powder (Merck) in MilliQ and autoclaving for 30 minutes. Concentrations





determined spectrophotometrically according to Hansen and Koroleff (1999). The detection limit ranged between 0.00 and 0.01 µmol L$^{-1}$ over the study period.

Subsamples for biogenic silicate (BSi) were also taken by vacuum filtration with the particles collected onto a cellulose acetate filter (0.65 µm pore size, Whatman). The filters were leached in NaOH (0.1 mol L$^{-1}$) at 85°C to dissolve the biogenic silicate. The leaching was terminated after 135 minutes by addition of acid (0.05 mol L$^{-1}$ H$_2$SO$_4$) and the dissolved silicate concentrations were determined spectrophotometrically according to Hansen and Koroleff (1999). The detection limit ranged between 0.01 and 0.36 µmol L$^{-1}$ over the study period.

### 2.4.4 Chlorophyll *a* concentration

Samples for pigment analysis were collected by vacuum filtration onto glass fibre filters (nominal pore size 0.7 µm, Whatman). Filtration volumes ranged between 1000 and 1500 mL. Filters were stored in cryovials at -80°C until extraction in acetone (100%, HPLC grade, Merck) as described in Paul et al. (2015). Analysis of Chlorophyll *a* concentration was carried out by High Performance Liquid Chromatography with concentrations calibrated against commercial pigment standards.

### 2.4 Data analysis

The experiment was separated into two time periods to facilitate interpretation of the system response: short-term (2-12 days after treatment, Days 7-19), longer term (>14 days after treatment, Days 21-33). This phase definition also enabled distinction between a period with stable total alkalinity and low Chlorophyll *a* (short-term), and a following period where in some mesocosms a significant bloom in Chlorophyll *a* occurred and a decrease in total alkalinity was measured (longer term). Each time period contained seven sampling days.

Simple linear regressions with Ocean Alkalinity Enhancement (OAE) or the measured total alkalinity as predictor variables were employed. The phase averages of the different system properties were the response variables. All data points that were considered analytically robust were used in the regression analysis to avoid a left-skewing of the data from zero inflation. This retained the relative treatment differences of interest, even if the absolute values are not realistic, for example negative nutrient concentrations. A few measurements provided unnatural values and were thus excluded. This outlier removal at the level of individual days and mesocosms did not compromise the integrity of the regression analysis conducted on the phase averages. Assumptions of normality of residuals and heteroscedasticity were assessed visually using residual and Q-Q plots. Data transformations were applied where necessary. All data analyses were performed at a significance level of α = 0.05 using R (version 3.6.3, R Core Team, 2020).





## 3 Results

### 3.1 Verification of alkalinity enhancement, shifts in seawater chemistry and carbon storage durability

The nine OAE levels ranged from ambient (~2400) up to ~4700 µeq kg$^{-1}$ and indicated successful implementation of the alkalinity gradient (Fig. 2a). Alkalinity enhancement created gradients in the speciation of the seawater carbonate system (Fig. 2c-f). Overall, doubling alkalinity led to ~2x higher bicarbonate concentrations, ~3x higher saturation state of aragonite ($\Omega_{Ar}$), and ~2x lower proton (H$^+$) concentrations (higher pH) than under natural alkalinity. Minimum $\Omega_{Ar}$ was 2.8 indicating supersaturation of aragonite ($\Omega_{Ar} >1$). Mean bicarbonate concentrations and $\Omega_{Ar}$ increased, and mean H$^+$ concentrations decreased linearly with increasing seawater alkalinity.

Total alkalinity and dissolved inorganic carbon concentrations (DIC) remained relatively constant in each mesocosm until Day 21. Thereafter a decrease in alkalinity and DIC was observed under the highest OAE, that accelerated in the loss rate over time (Fig. 2a, b). This amounted to a total alkalinity loss of 270 µeq kg$^{-1}$ and ~140 µmol kg$^{-1}$ dissolved inorganic carbon by Day 33, with an additional alkalinity loss of >350 µeq kg$^{-1}$ over 10 days between Day 35 and 45 (shown separately in Fig. 3b).






**Fig. 2: Carbonate system variables over time and in response to Ocean Alkalinity Enhancement OAE. a) Total Alkalinity (TA) and b) Dissolved Inorganic Carbon (DIC) under different levels of Ocean Alkalinity Enhancement (OAE). Panels c)-f) show average of key carbonate system variables in response to measured total alkalinity for the two response periods (short/longer-term).**



In conjunction with the decrease of alkalinity and DIC, precipitation of calcium carbonate occurred from Day 21 on. However,

the concentrations of dissolved $CO_2$ increased due to shifts in the carbonate system under lower alkalinity and unlike other carbonate system variables, did not display a linear relationship with alkalinity (Fig. 2f). The loss of alkalinity in the mesocosm OAE2400 had an impact on the carbon storage stability. Initially, DIC increased by ~1700 µmol kg$^{-1}$ via the addition of alkalinity on Day 7. However, a decline in DIC was visible after Day 23, and approximately 10% of the initially stored carbon was lost over the 25 days post-treatment (Day 7). The relative change in TA and DIC over time was calculated and compared

with salinity changes (~1 PSU) to account for evaporation that occurred during the study. No measurable change in carbon storage was observed at lower alkalinity that could be attributed to alkalinity loss (Fig. 2b). The ratio of TA to DIC loss was ~2.5 µeq:1 µmol and therefore above the ratio indicating ideal carbonate precipitation (2:1). The elevated $pCO_2$ of 541 µatm in OAE2400 on Day 33 indicates that gas exchange across the air-water interface in the mesocosm was slower than the shifts in the carbonate system induced by alkalinity loss. Hence the mesocosm became supersaturated in $pCO_2$ compared to the

atmosphere.

In the separate incubation study with the two highest OAE treatments (OAE2100, OAE2400), measured alkalinity concentrations decreased in all seeded treatments indicating that alkalinity loss was initiated by the calcium carbonate particles added, even at a lower alkalinity of OAE2100 (Fig. 3a). Alkalinity was stable in both unseeded treatments for both OAE levels

(OAE2100, OAE2400) using fresh seawater. Alkalinity declined in unseeded OAE2400 using mesocosm water and at a similar rate to the seeded treatments. This indicates that precipitation nuclei were already present in the OAE2400 mesocosm water but not in the fresh seawater. The data from the fresh seawater incubations is already presented in Hartman et al. (2023, see Fig. 4).



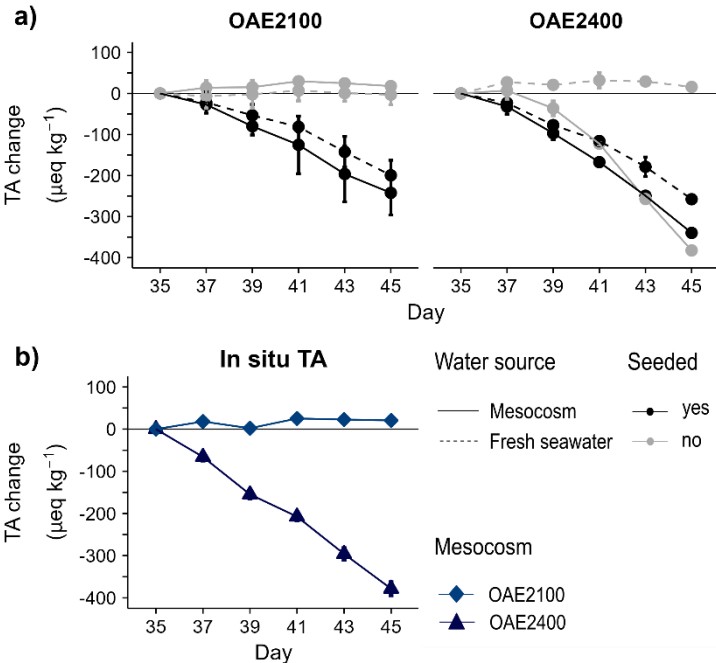

**Fig. 3: Change in seawater alkalinity (mean ± S.D., n = 2) between Days 35-45 a) in response to seeding with calcium carbonate particles at OAE2100 and OAE2400 in the incubation experiment and b) in situ in the two mesocosms with the same alkalinity treatment which were regularly sampled after the end of the main experiment. One replicate is missing from the seeded OAE2400 incubation on Days 41-45 so no error bars can be given for this treatment.**

### 3.2 Inorganic nutrients

There was no significant relationship between OAE and inorganic nutrient concentrations measured in either phase. $NO_x$ concentrations remained very low and mostly below analytical detection limits throughout the 33-day study period (Fig. 4a). Approximately 0.1 µmol $L^{-1}$ inorganic phosphate ($PO_4^{3-}$) was consumed before Day 13 with low and stable concentrations thereafter (Fig. 4b). Silicic acid concentrations were initially 0.42-0.44 µmol $L^{-1}$ (Fig. 4c) and consumption ranged from 0.07-0.22 µmol $L^{-1}$ between Day 3 and Day 19, with minimal change thereafter.




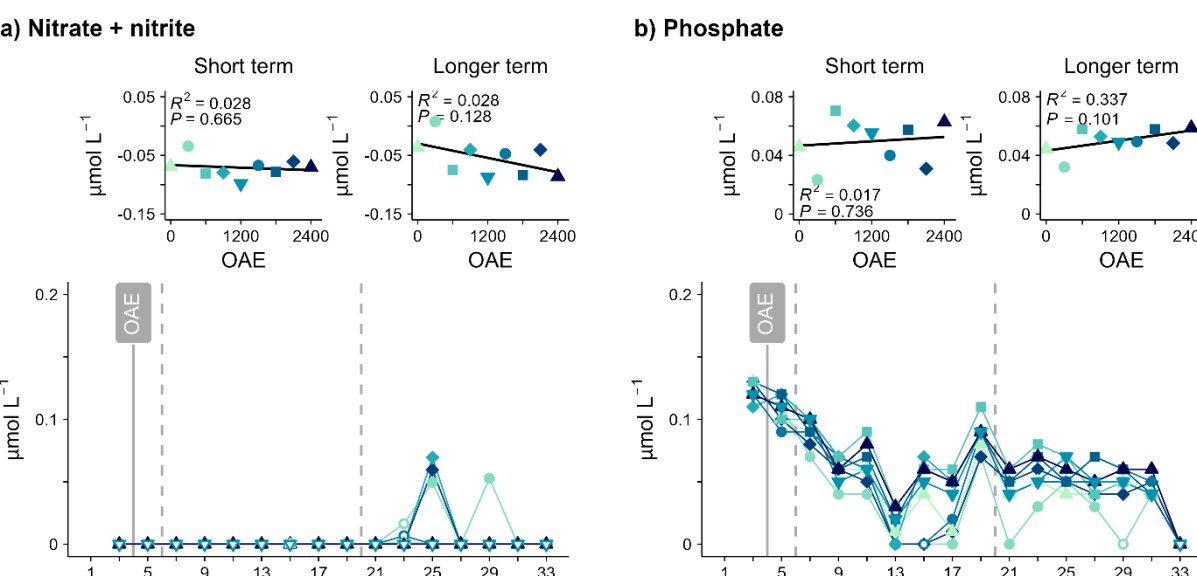

## c) Silicic acid

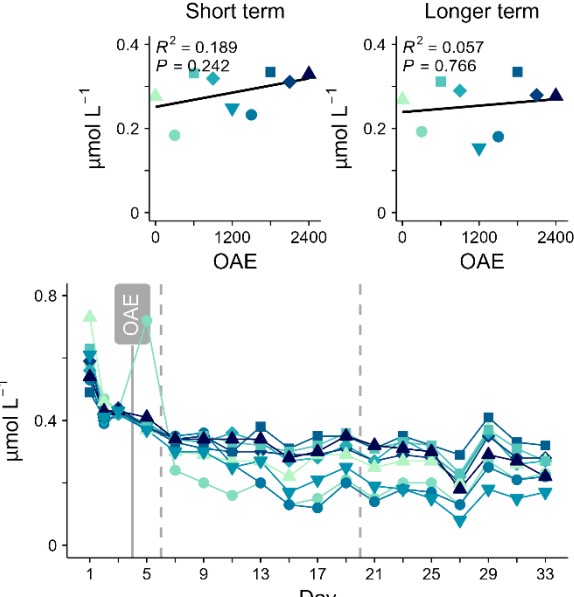

**Fig. 4: Nutrient concentrations over time and in response to Ocean Alkalinity Enhancement OAE. Open symbols indicate data points excluded from linear regressions.**



### 3.3 Particulate matter

No relationship between total Chlorophyll *a* (Chl *a*) concentration, a proxy for phytoplankton biomass, and OAE was detected in either response period. Chl *a* was below 0.5 µg L$^{-1}$, declined slightly in all mesocosms before alkalinity addition and remained low until Day 21 (Fig. 5a). After Day 21, Chl *a* increased in some mesocosms, reaching up to 5.5 µg L$^{-1}$ on Day 27 with a rapid decline in concentrations evident by the end of the study period. Bloom occurrence and magnitude was not related to OAE (Fig. 5c) and occurred despite very low nitrate concentrations.


No evidence for a relationship was detected between OAE and suspended particle organic carbon (POC) or phosphorus (POP) concentrations in any phase. However, average PON concentrations were 19 % lower in the highest OAE treatment compared to ambient before Day 19 (Fig. 5, Table S1). A similar (negative) relationship was also seen in POC and potentially in POP concentrations but this was not statistically significant. This would indicate a general response in this experiment that less

particulate organic matter was retained in the water column under enhanced alkalinity, even though it was only the PON pool where this impact was clear. This impact of OAE was only transiently measured in the PON pool and the longer term OAE response was not significant (Table S1).

After Day 21, bloom development unrelated to alkalinity appeared to be the dominant driver of variability in the particulate

matter concentrations (POC, PON, POP). POC, PON, and POP followed a similar temporal pattern to Chl *a* concentrations, with relatively stable measured concentrations observed until Day 21 and a noticeable increase in POC, PON and POP concentrations in the mesocosms with the highest Chl *a* concentrations after Day 21 (Fig. 5).

Mean POC:PON was positively correlated with OAE between Days 21-33 (Fig. 6, Table S1), indicating particles were

generally richer in carbon under higher alkalinity. No linear relationship between PON:POP or POC:POP and OAE could be detected in either phase. PON:POP and POC:POP showed no clear temporal trends but were more variable over time compared to POC:PON, and indicated relatively phosphorus poor organic matter compared to the Redfield ratio (N:P ~16, C:P ~106, Fig. 6). POC:PON was relatively constant throughout the study period with values around or above the Redfield ratio (C:N = 6.6, Fig. 6a). After Day 21, PON:POP and POC:POP was highest in the treatments which also had the highest Chl *a*

concentrations (OAE1500, OAE 1800, Figs. 5, 6).



**Fig. 5: Particulate matter concentrations over time and in response to OAE. Open symbols indicate data points excluded from linear regressions. Note that Chlorophyll *a* y-axis is on a logarithmic scale.**





**Fig. 6: a)-c) Particulate matter stoichiometry and d) stable isotope abundance in particulate nitrogen (δ¹⁵N) over time and in response to Ocean Alkalinity Enhancement (OAE). Horizontal dashed lines indicate the Redfield ratio for each element pair (C:N = 6.6, N:P = 16, C:P = 106).**



### 3.5 Biomineralization indicators

Higher PIC:POC with higher alkalinity was observed before Day 19. Other indicators of calcification (PIC, particulate inorganic carbon) and silicification (BSi, biogenic silica; POC:BSi), did not indicate any significant impact of alkalinity (Fig. 7, Table S1). There was also no clear increase in either BSi or PIC concentration around Day 21-27 in the mesocosms where the bloom developed, and where the shifts in other particulate matter variables or Chl *a* were observed.

Biogenic silicate (BSi) concentrations were initially below the analytical detection limits, briefly increasing to up to 0.25 µmol L$^{-1}$ until around Day 7, before declining and reaching a minimum around Day 19-23 (Fig.7a). This decline in BSi concentrations was evident in the POC:BSi stoichiometry during the same period to above 800 mol:mol in some treatments, and suggests that the contribution of silicifying phytoplankton to organic matter production was particularly low during this period. Particulate inorganic carbon (PIC) concentrations were very low throughout the study period and not detectable (<0 µmol L$^{-1}$) on many days (Fig. 7c). This was due to the measured total particulate carbon (TPC) content being lower than the organic particulate carbon (POC) content for the same treatment, a result of method-related variability. PIC:POC was also usually low (<0.5 mol:mol) and there were no consistent or clear changes over time. Both PIC concentrations and PIC:POC indicate low calcification. SEM photos in addition to FT-NIR/EDX analyses of inorganic particles collected from the mesocosm walls of OAE2400 (Fig. S2) indicate that this was not biologically controlled carbonate precipitation of aragonite (Fig. S3).

### 3.6 δ$^{15}$N-PN

There was no relationship between stable isotope abundance in particulate nitrogen (δ$^{15}$N-PN) and alkalinity. δ$^{15}$N-PN was initially around 5 ‰ and similar across all mesocosms until around Day 21 (Fig. 6d), indicating nitrogen cycling was initially similar across all treatments. Thereafter a divergence between mesocosms emerged that was sustained until the end of the study period (Day 33). A decrease was particularly evident in the mesocosms where the phytoplankton bloom was observed. The mesocosms with the lower Chl *a* concentration did not show this same decline and δ$^{15}$N-PN was even slightly higher than those after Day 21. These two observations could indicate that a different N source was used over time and between the blooming and non-blooming mesocosms.





**Fig. 7: a)-b) Indicators of silicification (BSi, POC:BSi) and c)-d) calcification (PIC, PIC:POC). PIC:POC was sometimes below 0 and are not shown on the temporal plot but all data are included for the regression plots.**




## 4 Discussion

**4.1 Doubling of seawater alkalinity had minor impacts on biogeochemistry in low nutrient ecosystem**

The detection of only 3 significant differences in biogeochemical pools suggests that the risk of major changes to biogeochemical functioning for the applied OAE scenario is low (Table 1). Our study in a low nutrient plankton assemblage found only a minor impact of carbonate-based, $CO_2$-equilibrated ocean alkalinity enhancement (OAE) on a suite of biogeochemical properties that relate to nutrient utilisation and particulate matter production and stoichiometry. These findings 365 align with those by Subhas et al. (2022) who studied microbial communities in the North Atlantic Gyre.

| Associated ecosystem function | Measured biogeochemical property | | Impact of OAE | |
|---|---|---|---|---|
| | | | *Short term* | *Longer term* |
| • Nutrient cycling | Inorganic nutrients | nitrate | = | = |
| | | phosphate | = | = |
| | | silicate | = | = |
| • Nutrient cycling | Particulate matter concentration | Phytoplankton biomass (Chl *a*) | = | = |
| • Primary production | | carbon (organic, POC) | = | = |
| • Carbon storage | | **nitrogen (PON)** | ↓ | = |
| • Food quantity for consumers | | phosphorus (POP) | = | = |
| | | biogenic silica (BSi) | = | = |
| • Nutrient cycling | Particulate matter stoichiometry | **POC:PON** | = | ↑ |
| • Carbon storage | | POC:POP | = | = |
| • Consumer food quality | | PON:POP | = | = |
| | | PON:BSi | = | = |
| • Alkalinity loss* | Biomineralisation indicators | carbon (inorganic, PIC) | = | = |
| • Carbon sequestration | | **PIC:POC** | ↑ | = |
| | | POC:BSi | = | = |

**Table 1: Impact of alkalinity enhancement on measured biogeochemical variables. "=" indicates no effect detected. * not an ecosystem function but is important in determining the efficacy of OAE.**

We attribute the biotic response stability in part to the low baseline nutrient concentrations in both studies. Low nitrate concentrations mean there was limited potential for substantial bloom development and emergence of treatment differences in relevant dissolved and particulate matter pools. Primary producers were limited in inorganic nutrients but they were not





subjected to carbon limitation, a consequence of the $CO_2$-equilibrated alkalinity addition in this particular study. Hence, when nutrients are the ultimate limiting factor, the response to other drivers like carbonate chemistry may not be visible. Here,

heterotrophy drove the observed ecosystem responses. In nutrient-rich ecosystems, dissolved $CO_2$ supply may instead limit or co-limit phytoplankton growth (e.g. Riebesell et al., 1993). Consequently, the response in biogeochemical variables to drivers such as OAE could be different, also in magnitude and direction, as has been observed in ocean acidification studies (Sala et al., 2016; Taucher et al., 2018). This is particularly likely in a non-equilibrated alkalinity enhancement scenario and more productive ecosystems, where dissolved $CO_2$ concentrations may be further reduced through high consumption in bloom

periods. Nevertheless, even under nutrient replete conditions, heterotrophic processes in a microbial food web seem to be even more sensitive to ocean alkalinity enhancement than autotrophic production (Ferderer et al., 2022). Similar to our findings, heterotrophy also shaped the stoichiometric response in particulate organic matter to enhanced alkalinity.

One particularly intriguing outcome was that two observed impacts on organic matter were related to nitrogen pools (PON,

POC:PON) in this N limited plankton community. Neither POP nor N:P or C:P were affected by OAE within the study period, indicating distinct impacts on N turnover relative to C and P turnover in this plankton community related to carbonate system changes. Many microbial N-cycle processes such as ammonia oxidation, nitrification and urease activity are pH (i.e. proton-concentration) dependent (Beman et al., 2011; Fumasoli et al., 2017; Pommerening-Röser and Koops, 2005). Hence, this leads us to the conclusion that heterotrophic turnover of organic nitrogen may be particularly influenced by OAE, although observed

optimal pH usually lie below the modest pH range implemented in this study.

Ecologically, this measured decrease in particle nitrogen richness would mean a decline in food quality for higher trophic levels (Anderson et al., 2005), which depend on the organic matter created for energy and nutrient provision. This would be a negative consequence of enhancing alkalinity in this marine ecosystem. Due to the small initial response of C:N to OAE and

overall low productivity, the corresponding food web impact would be also minor, hence it is unlikely that this impact could have been detected in food web related variables such as trophic transfer efficiency. Biogeochemically, an increase in C:N in the suspended particulate material is a potential additional carbon sequestration process. Higher C:N is essentially enhancing the organically bound carbon content of particles which may then sink out of the surface ocean via the biological carbon pump. This would be a co-benefit of OAE and increase the carbon sequestration efficiency.

**4.2 Alkalinity enhancement efficacy and durability**

Our monitoring of the alkalinity pool indicated that carbon sequestration efficacy even for the most conservative approach of enhancing alkalinity - dissolved and equilibrated with $CO_2$ – is not necessarily durable, if certain thresholds are exceeded (Moras et al., 2022, Hartmann et al., 2023). Doubling alkalinity was initially effective in maintaining the amount of carbon chemically bound. However, slow but accelerating alkalinity leakage was observed from within two weeks following doubling

of seawater alkalinity. Hence our study shows that the runaway process can also occur outside of a lab bottle experiment in a





close-to-natural system. This leakage corresponded to a total 10 % decrease in carbon sequestration efficacy in the highest OAE treatment just 25 days after application. Results from the additional incubation experiment suggests that the loss of TA and DIC continued for until at least Day 45. Sporadic continued measurements indicating this loss continued for more than 90 days after the end of the study but does lead to complete removal of alkalinity (Fig. S4, Supplementary Material). Alkalinity
in incubations with mesocosm water seemed to reach a chemical threshold of $\Omega_{aragonite}$ that ranged between of 4 to 6. This corresponds to a loss of around 800-900 µeq kg$^{-1}$ TA (Fig. S4, Supplementary Material). Such behaviour would be consistent with further incubation side experiments conducted during the mesocosm experiments reported in Hartmann et al. (2023).

Calcium carbonate precipitation is the likely process behind this measurable loss in water column alkalinity and was confirmed
in incubation side experiments with TA:DIC loss ratios of about 2 and examining precipitates (Hartmann et al., 2023). Analysis of precipitates from the mesocosm experiment confirmed these were composed of calcium carbonate, and likely aragonite. While no loss in alkalinity could be directly quantified in other mesocosms in this study, more sensitive pools (PIC:POC in suspended and sinking particles) indicated that particle inorganic carbon content, hence carbonate precipitation, was higher with increasing alkalinity before Day 21. This precipitation may have even occurred at lower OAE based on collected sinking
particle carbonate content in the sediment trap (Suessle et al., n.d.), even though there was no measurable loss of alkalinity in the water column. Calcifying organisms are scarce in nutrient-poor seasons around the Canary Islands (Sprengel et al., 2002) and were not in high abundance in this experiment either (X. Xin/A. Stuhr, pers. communication). Precipitation of carbonate is reported during alkalinity enhancement in seawater (Griffioen, 2017; Moras et al., 2022; Subhas et al., 2022) and can be initiated by the presence of particles that act as nuclei (Hartmann et al., 2023; Wurgaft et al., 2021). Our separate short-term
incubations of mesocosm water showed that sufficient nuclei were already present in OAE2400 to sustain the precipitation and alkalinity loss we had directly observed in the mesocosm itself. At slightly lower enhanced alkalinity (OAE2100), no decline in alkalinity was observed until precipitation nuclei were added. Hence, significant alkalinity loss appears to depend on the generation or presence of precipitation nuclei, which we only observed in the water column above an apparent threshold OAE 2100 over the limited time period in our particular experiment.


The secondary precipitation appears to be primarily controlled by abiotic conditions, the saturation state, as indicated by side experiments (Hartmann et al., 2023) but does seem to have originated in a benthic film in OAE2400, created either abiotically, or related passively or actively to microbial activity (Dupraz et al., 2009) under the higher seawater saturation state. We regularly wiped the inside of the mesocosms to remove growing microbial biofilms from the submerged parts of the mesocosm
walls. However, white particles were visibly attached to the mesocosm wall material around Day 28 (see photo in Fig. S2), developing rapidly in the 6 days after the previous cleaning. Cleaning procedures then simply removed the precipitated calcium carbonates from the walls and transformed these to particles suspended and sinking in the water column. This primarily benthic origin is in contrast to previous studies where the precipitation nuclei started in the water column as suspended particles (Moras et al., 2022). For ocean alkalinity enhancement implementation, this passive, but biotically-related, precipitation (see e.g.



Castanier et al., 2000; Hammes and Verstraete, 2002; Novitsky, 1983) on surface biofilms would present more of an issue for nuclei generation in coastal rather than open ocean OAE, when alkalinity is enhanced in a dissolved form. Shallow coastal ecosystems have large benthic substrate surface areas where biofilms develop rapidly in the nutrient- and light-rich environment and could potentially influence precipitation nucleation and alkalinity loss, as we suggest occurred in our study.

### 4.3 Substantial bloom development under nitrate-limitation with enhanced alkalinity

We observed a substantial phytoplankton bloom (Chl *a*) in selected mesocosms that was clearly decoupled from inorganic nutrient concentrations (N, P). However, is not clear why this bloom only occurred in some mesocosms. This could be a purely coincidental observation as there was no clear linear relationship between Chl *a* concentrations and alkalinity treatment. Nevertheless, this bloom only occurred in mesocosms where alkalinity was increased and the carbonate system shifted (higher bicarbonate and carbonate, lower H$^+$).


An assessment of previous mesocosm studies carried out in the same ocean region (Fig. 8) implies that this nutrient-decoupled bloom is a novel response. Such a substantial bloom has not been seen in previous nutrient-deplete experiments, or experimental phases simulating ocean acidification and where seawater carbonate chemistry (pH, pCO$_2$) was modified (e.g. Taucher et al. 2018, KOSMOS2016/2017: unpublished, see Supplementary Material, Table S3). Indeed, the bloom magnitude

of over 4 µg Chl *a* L$^{-1}$ was similar to that arising from addition of nutrient-rich deep water (Taucher et al., 2018).

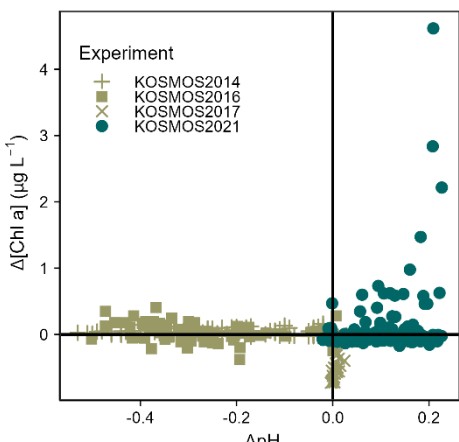

**Fig. 8: Difference in Chlorophyll *a* concentrations compared to difference in seawater pH from four mesocosm experiments in the North Atlantic Ocean, calculated compared to the control mesocosm on Day 1. Each point corresponds to one value from one**
**mesocosm on one sampling day. Further information on experiments are provided in Supplementary Material (Table S3).**

It is still uncertain if this bloom was due to the increase in alkalinity and to what degree production (i.e. bottom-up) processes or grazing influenced these bloom dynamics. However, we identified two mechanisms related to nutrient availability, which could explain this unusual bloom occurrence and provide a potential link to OAE. Firstly, plausible direct links exist between



bicarbonate and carbonate ion concentration and the ability of phytoplankton to access trace metals or nutrients from the
microenvironment surrounding the cell (Coale et al., 2019; Liu et al., 2022; McQuaid et al., 2018). Hence, chemical
perturbations from OAE have potential to modulate microbial nutrient availability and thereby enable such blooms in nutrient-
poor ecosystems. Secondly, nutrient availability for primary producers could be modified indirectly through the heterotroph
response to OAE. Here, this could be related to microbial turnover of organic nitrogen. Impacts of OAE were most clearly
observed in nitrogen-related pools hence indicating heterotrophs were more sensitive than autotrophs to OAE (see Sect. 4.1).
Furthermore, lower $^{15}$N enrichment of particulate nitrogen in the blooming mesocosms points towards utilisation of a different
nitrogen source by primary producers in these mesocosms, for example, ammonium regeneration (Mercado et al., 2010) or $N_2$
fixation. Although these observations did not necessarily occur in the same mesocosm, organic nitrogen metabolism and
passive $CaCO_3$ precipitation can be related processes in microbes (Achal and Pan, 2011; Vincent et al., 2022). This suggests
that the response of heterotrophs to OAE may also compromise the stability of an alkalinity addition. Thus, if this unusual
bloom is a consistent response to increased alkalinity, it will be important to probe the underlying mechanisms in depth because
this indicates that a side effect of alkalinity enhancement could be increased ecosystem productivity and relief of nutrient
limitation in highly oligotrophic regions, and may impact the efficiency of OAE implementation.

## 5 Conclusions

Our study simulated an idealised OAE deployment (carbonate-based, $CO_2$-equilibrated) in a natural plankton community.
Seawater alkalinity was modified without addition of trace metals or other ions that occur in olivine addition or other enhanced
mineral weathering approaches and the $CO_2$ was already chemically sequestered. Overall our results suggest that even with a
doubling of seawater alkalinity, there is a low risk of major changes in biogeochemical functioning. This gives greater
confidence that the alkalinity perturbation itself will not be the most critical factor when considering equilibrated ocean
alkalinity enhancement as a carbon dioxide removal approach. However, our study demonstrates a risk that added alkalinity
may be lost above a threshold of OAE. Hence, durability of the chemical $CO_2$ sequestration may be constrained by the degree
of alkalinity enhancement implemented due to generation of particles that can nucleate carbonate precipitation. The higher the
perturbation, the higher the risk that precipitation at nuclei initiates and this rapid leakage process develops. It is also not yet
clear if coastal environments with high surface areas and calcium carbonate structures may also facilitate precipitation.

Under nitrate limited growth in primary producers, nitrogen-related pools were two of three pools affected in this mesocosm
study. Substantial growth in Chl $a$-containing organisms under nitrate-limitation also only occurred in mesocosms where
alkalinity was added. Both observations suggest some influence of enhanced alkalinity on nutrient turnover and nitrogen-cycle
processes, in addition to the accumulation of carbon-richer particles under higher alkalinity. These warrant closer inspection
in future studies to clarify the magnitude of the processes and the interaction with this novel potential perturbation to the
carbonate system in large ocean areas with low inorganic nitrogen concentrations.



**Data availability**

All data have been submitted for open access on the PANGAEA data portal and the doi are currently under assignment for the following data sets

Mesocosm study:

Incubation study and extended measurements:

**Supplement link**

(to be added by Copernicus)

Table S1: Overview on linear regression analyses

Table S2: Carbonate chemistry treatment and phase means

Table S3: Overview of data included in Fig. 8 in manuscript

Figure S1: Photo of mesocosm set up at pier

Figure S2: Photo of white particles attached to mesocosm walls in OAE2400

Figure S3: FTIR spectrum of mesocosm precipitates and calcite/aragonite standard materials

Figure S4: Long-term changes in alkalinity (TA) and aragonite saturation state ($\Omega_{ar}$) from extended incubations

**Author contribution statement**

AP, MH, NSa, NSu, SG, and UR designed and conceptualised the mesocosm study with NSa, MH and SG collecting and analysing samples in the laboratory, JS and NSu carried out sampling and laboratory analysis for the incubation study. AP,

JH, JS, MH, SG, NSa, NSu, and UR analysed and interpreted the data with AP preparing the manuscript with contributions from all co-authors.

**Competing interests**

Allanah J. Paul has been employed by the non-profit organisation Bellona as a CDR Research and Technology Advisor since October 2023. The research reported in this manuscript was completed prior to starting this role. Allanah is also an external

scientific advisor to "Seafields" (https://www.seafields.eco/), an aquaculture business for CDR using seaweed.



**Special Issue Statement**

(to be added by Copernicus)

**Acknowledgements**

The authors would like to thank the Oceanic Platform of the Canary Islands (PLOCAN) and its staff for the use of their marine
and terrestrial facilities, and for their help with the logistics and organisation of this experiment. This study involved a large
team effort and we thank all study participants for their contributions on site in Taliarte, Gran Canaria. In particular we thank:
Andrea Ludwig and Jana Meyer for logistical support and coordination of on-site activity; Anton Theileis, Jan Hennke and
Michael Krudewig for mesocosm preparation, technical support and maintenance; Daniel Brüggemann, Philipp Süßle, Joaquin
Ortiz, Carsten Spisla and Michael Sswat for onsite scientific diving activities and maintenance; Kerstin Nachtigall, Levka
Hansen, Anna Groen, Jannis Hümmling, Jana Willm, and Juliane Tammen for laboratory support in Taliarte and in Kiel . We
also thank Javier Aristegui and Laura Marin-Samper (ULPGC) for interesting discussions around the observed phytoplankton
bloom, and to the KOSMOS2014, 2016 and 2017 teams and students for the data presented in Fig. 8. This study was funded
by the OceanNETS project ("Ocean-based Negative Emissions Technologies – analysing the feasibility, risks and cobenefits
of ocean-based negative emission technologies for stabilizing the climate", EU Horizon 2020 Research and Innovation
Programme Grant Agreement No.: 869357), and the Helmholtz European Partnering project Ocean-CDR ("Ocean-based
carbon dioxide removal strategies", Project No.: PIE-0021) with additional support from the AQUACOSM project (EU
H2020-INFRAIA Project No.: 731065, "AQUACOSM: Network of Leading European AQUAtic MesoCOSM Facilities
Connecting Mountains to Oceans from the Arctic to the Mediterranean").

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
