# Peer review of "Ocean alkalinity enhancement in an open ocean ecosystem: Biogeochemical responses and carbon storage durability"

_EGUsphere, 2024_

## Author Response (AR1)

**Response to comments from Reviewer 1**

We are very happy to receive the positive comments from Nicholas Ward and thank them for their constructive feedback on how to improve the manuscript. We have responded to each comment below in italics.

*We have noted the specific changes to the manuscript here in italics below our previous comments. The line number reflect the revised manuscript version.*

**General Comments:**

Paul et al., describe a very interesting set of in situ OAE mesocosm experiments with varying levels of carbonate alkalinity added to large floating chambers. The study evaluates an impressive suite of bulk biogeochemical parameters allowing the authors to examine responses of multiple coupled carbon and nutrient cycle behaviors. The experiments were well designed to address the objective of the study – understanding whole system response to alkalinity addition.

The results show that fairly high alkalinity additions (up to doubling) have fairly minimal impacts on other biogeochemical cycles other than some unexpected bloom behaviors, which could just as likely be related to random mesocosm related issues. The main findings of note are that 10% of the alkalinity was lost to precipitation in the highest treatment and the carbon/nitrogen content of particles slightly shifted, though the mechanism for the latter is uncertain.

Overall this is a solid assessment of OAE impacts and a very useful body of work for this emerging field. I am supportive of this study's publication and have mostly minor comments below.

**Specific Comments:**

Line 24: Consider making this statement quantitative, e.g., "X% of the added alkalinity was lost"

Abstract in general: The findings are described fairly generally and there's little to any quantitative statements or comparisons. A lot of parameters were measured in this study, so I understand there is a lot to summarize. But after reading the abstract I only understand very high level observations. Adding quantitative elements to this summary would be useful, e.g. how much lower was PON in the high alkalinity treatment, how much alkalinity was lost (and perhaps what was the particle load that allowed nucleation to occur), etc.?

Author response: We thank the reviewer for raising this in these two comments about providing quantitative statements. This was a matter of discussion between the co-authors during manuscript editing. We do include quantitative information on loss of initially stored carbon in the results section in the manuscript, where there is space to provide the necessary context of these results. However, between the co-authors we decided against including quantitative statements in the abstract because this is highly dependent on the experimental conditions and also to the observed time periods. Unfortunately, we do not have any quantitative information

on particle load or concentration, but we will reconsider which comparisons and quantitative statements we can add to the abstract in revising the manuscript.

*Author response: We have made the following change to lines 25-26 in the revised manuscript (new text is underlined):"The loss rate in this mesocosm accelerated over time and amounted to ~10% of stored $CO_2$ within 4 weeks after alkalinity enhancement."*

Line 51: Define NETs

Author response: NETs refers to "Negative Emission Technologies". We will add this information to line 51.

*Author response: This information had been added to line 53 in the revised manuscript.*

Line 138: Can you give more details on the isotope tracer? E.g., what percentage 13C was the compound and what was the compound (e.g., CO2, HCO3, CO3)? Also instead of stating trace amounts, be specific about how much was added.

Author response: The isotope tracer added was $^{13}C$-NaHCO$_3$. We added 10g of $^{13}C$-bicarbonate on Day 11, enriching the isotopic signature of the inorganic carbon pool by 100‰ (in terms of %At $^{13}C$, it went from 1.105±0.001 (baseline) up to 1.193±0.001 (based on measurements on day 13). Further details are reported in Sanchez et al. (in review), along with the results on the transfer of carbon throughout the food web that was behind the use of this tracer. We reported this detail on $^{13}C$ here for completeness of all study manipulations but will remove unnecessary details for this manuscript in the revised version.

*Author response: This information has been removed from lines 143-144 in the revised manuscript and the detail on 13C addition has been removed from Fig. 1 (Experimental timeline). These changes have been made in line with comments from Reviewer 2.*

Line 140: Were the incubations at the end exposed to light or done in the dark?

Author response: The incubations carried out at the end of the mesocosm study (Days 35-45) were carried out in submerged bottles in Taliarte Harbour, where they were exposed to natural light (see line 162).

*Author response: No additional information added as information is already reported in the manuscript.*

Line 234: Can you explain this approach a bit more? I'm not following why you might have negative nutrient values, is this related to the data analysis approach or a negative value reported by an instrument (i.e., sample below the detection limit)?

Author response: This approach of taking the absolute nutrient concentrations for the statistical analysis, was used to avoid issues of zero inflation that may have biased the results in the data analysis. This was because of an issue with the blanks in the calibration that had higher nutrient concentrations than some of the samples of the oligotrophic water enclosed in the mesocosm. The relative differences are real, even if the absolute values are not.

*Author response: This information has been added to line 243 in the revised manuscript to read: "This retained the relative treatment differences of interest, even if the absolute values are not realistic, for example negative nutrient concentrations due to issues with the blank values in the calibration."*

Line 277: Is this sentence suggesting that the seawater used in the 2400 experiment may have had different particle loads than the other mesocosms? Or do you think 2400 just happened to be some important threshold in aragonite saturation in relation to the ambient particle load?

Author response: The initial water enclosed in the mesocosms was the same, so the precipitation nuclei must have been produced in the 2400 OAE treatment during the study period and after the OAE treatment was applied. We tend to think that the 2400 OAE treatment had an aragonite saturation level that led to production of precipitation nuclei, although we don't have any clear evidence to support this. We do note that the saturation state of aragonite was very close to the theoretical threshold for homogeneous carbonate precipitation suggested by Marion et al. 2009 (https://doi.org/10.5194/os-5-285-2009), based on Morse and He 1993 (https://doi.org/10.1016/0304-4203(93)90261-L) and Morse et al. 2007 (https://doi.org/10.1021/cr050358j).

*Author response: Information has been added to lines 282-289 in the revised manuscript (new text underlined): "As the initial water enclosed in the mesocosms on Day 0 was the same, the precipitation nuclei must have been produced in the 2400 OAE treatment at some point during the mesocosm study, and related to OAE2400 treatment application. Alkalinity was stable in both unseeded treatments for both OAE levels (OAE2100, OAE2400) using fresh seawater. Alkalinity declined in unseeded OAE2400 using mesocosm water at a similar rate to the seeded treatments. This indicates that precipitation nuclei were already present in the OAE2400 mesocosm water but not in the fresh seawater. We note that the saturation state of aragonite was very close to the theoretical threshold for homogeneous carbonate precipitation suggested by Marion et al. (2009)), based on Morse and He (1993) and Morse et al. (2007)."*

Line 314: Can you use some actual values in this section? For example, "POC:PON increased from X to Y across the OAE treatments." Also, note in this section that this shift appears to be more related to a decrease in PON as opposed to increase in POC.

Author response: We will add some values from the statistical analysis to the text, such as the example given for the higher observed POC:PON with higher OAE treatment. We do mention that PON is lower under OAE on lines 302-303, however this was only observed as a transient response that was not observed longer term (after Day 19). Hence, we would be careful about drawing too strong a link between these observations.

*Author response: We have added the % difference in POC:PON (20%) to line 326.*

Line 335/Figure 7: For the biogeonic silica data it seems like data that's below the instrument detection limit needs to be flagged in the figure if it's being shown. A value below the detection limit means you can't confidently say the value is a real number above zero.

Author response: Thank you for picking up on this oversight. We will make sure all values below the detection limit are indicated as such in Fig. 7, as we agree that we cannot be confident about concentrations below the detection limit.

*Author response: We have modified Fig. 7 accordingly.*

Results in general: I don't see any results related to the 13C tracer. Why was it mentioned in the methods if it's not presented in the results?

Author response: It is correct that there are no results presented on the $^{13}$C tracer addition. These data are reported in the manuscript by Sanchez et al. (in review). We presented this in the experimental set-up for completeness, as this was an experimental manipulation that we thought was important to indicate.

*Author response: All references to 13C tracer addition have been removed from the revised manuscript (see also our response to Reveiwer 2 comments).*

Line 397: I understand this argument, but is it actually applicable to these results? The actual carbon content by mass didn't seem to go up, there was just less nitrogen driving this shift. If molecular composition were analyzed perhaps the case could be made that alkalinity enhancement somehow resulted in the production/presence of more refractory carbon compounds, though I'm not sure what the mechanism of this would be.

Author response: Yes, it is correct that the carbon content did not increase, and that this trend appears to have been driven by lower particulate organic nitrogen concentrations. However, we still think that this argument is valid because we are referring to the potential increase in carbon sequestration efficiency (relative change in C:N ratio) and not carbon pump strength, which would depend more on the overall amount of sinking, and potentially exported, material. Some more specialised and molecular analyses of particulate organic matter may have been able to detect any shifts in the presence of more refractory carbon in these particles, but unfortunately this was not part of the suite of measurements made. It could be an interesting follow-up analysis to consider in future studies.

*Author response: No changes made.*

Line 406: Can any statements be made about the physics of this study region? For example, is the flushing/residence time of Taliarte Harbour known, and would you expect that alkalinity additions would very quickly become diluted if performed here? I.e., would you ever expect to be able to achieve a +2400 alkalinity scenario that could yield similar precipitation in the environment?

Author response: We are not aware of any robust data sets on Taliarte Harbour itself, but the more relevant question would be for the coastal waters off Gran Canaria. There are strong currents in the area so an alkalinity addition, such as +2400 used in this study, would not be maintained for long. This study aimed to understand the general response of a subtropical ecosystem but we can imagine that the Canary Islands is representative of a highly flushed region where dilution occurs rapidly.

*Author response: We have added the following to lines 435-436 (new text underlined): "Precipitation of carbonate is reported during alkalinity enhancement in seawater (Griffioen, 2017; Moras et al., 2022; Subhas et al., 2022) and can be initiated by the presence of particles that act as nuclei (Hartmann et al., 2023; Wurgaft et al., 2021), and circumvented by rapid dilution after alkalinity increase (Suitner et al. 2024)."*

Line 473-477: This is all very speculative and cherry picking observations from different mesocosms. "This suggests" seems like too definitive of a statement. It's ok to speculate, but perhaps don't tie this discussion point as much to your experimental findings.

Author response: Thank you for picking up on this. We can see this was too definitive considering the lack of solid data to support this speculation. We will revise "suggest" to be less definitive such as "This may indicate that the response of heterotrophs to OAE may be relevant for the stability of an alkalinity addition".

*Author response: Sentence suggested above added to 486-487 in the revised manuscript.*

Line 485: Perhaps clarify that the threshold isn't just related to amount of OAE, the threshold seems to be related to both amount of OAE and in situ particle loads.

Author response: We agree that in situ particle loads may affect the stability of OAE but as we do not know the nature of the precipitation nuclei that seemed to be produced within the mesocosm system, we can only speculate on particle load. We will add to line 486 "... due to the generation, or natural presence, of particles that can nucleate carbonate precipitation", to indicate that particle load may be important.

*Author response: Suggested addition made to line 499 in the revised manuscript.*

**Review of Paul et al. "Ocean Alkalinity Enhancement in an Open Ocean Ecosystem: Biogeochemical Responses and Carbon Storage Durability"**

We thank the reviewer for their suggestions and constructive feedback on how to improve the manuscript. It gave us many interesting points to ponder and hope we have addressed these adequately in our responses (see below in italics).

*We have noted the specific changes to the manuscript here in italics below our previous comments. The line number reflect the revised manuscript version.*

This study evaluates the changes in inorganic carbon/ nutrients, and organic carbon /nitrogen over a month-long mesocosm experiment. They reported two very interesting findings are: a). DIC loss due to secondary precipitation. 2) Sensitivity of only organic nitrogen to OAE. It was a joy to read this paper. My comments are meant to challenge the authors to add more quantitative or semi-quantitative details about the mass balance instead of the mesocosm bags:

1. Nutrient mass balance. I am less convinced by the proposed nutrient sources (microbial nutrient viability and/or microbial turnover of organic nitrogen). If either of these processes were the cause, the PON and DIN pools should behave like a see-saw. However, Figures 4a and 5c show that both DIN and PON increased. Was there any external nutrient input during this period? DIP and POP seem to follow the see-saw pattern (though it is challenging to discern due to the similar colors in Figures 4 and 5). Can you examine the relationship between inorganic nutrients and particular organic species in each mesocosm to see whether there were any internal exchanges under OAE treatment?

Author response: It would be a huge step forward to be able to complete a mass balance for C/N/P/Si and other elements as this could be very informative on changes that OAE or other factors induce on biogeochemical cycles and marine productivity. Unfortunately, it is difficult to achieve this with reasonable precision in these mesocosms, due to the large size difference of the pools. The random measurement error in the larger pools (e.g. DON and PON) would overshadow any real changes in the smaller pool without the use of more precise isotope tracers. Our standing stock measurements are inappropriate to assess flows between pools and we did not take the necessary rate measurements that would be needed for this.

*Author response: No changes made. We would like to reiterate that DIN concentrations were almost always below detection (please see Fig. 4a).*

2. Additionally, BSi decreased over time in all treatments (Figure 7a), but silicic acid either stayed stable or slightly decreased. Can you explain the loss of Si in the mesocosm experiments?

Author response: Yes, this is because particles sink out of the water column over time and collect in the sediment trap. This means there is a net loss of particulate matter (containing C, N, P, Si, and other elements) from the mesocosms over time and there is a reduced opportunity for slowly remineralised nutrients like Si to return to the water column. This effect is more pronounced in more nutrient rich and productive ecosystems. In this low nutrient experiment, diatoms were not abundant and so silicic acid concentrations were relatively stable throughout the study period.

*Author response: We added information to lines 141-144 in the revised manuscript as follows:
"Biofilm material removed during mesocosm cleaning remains in the enclosed mesocosm ,either suspended in the water column or collected in the sediment trap. In both cases, this material is included in particulate matter analyses. Sediment trap material procedures and analyses are reported in Süßle et al.".*

3. Can you also clarify what happened to the biofilm after regular cleaning? I didn't find any related information in the manuscript. Did you remove the biofilm from the bags or discard it inside the bags? Did your POC, PON, and POP measurements include the impact of the biofilm treatment?

Author response: The cleaning procedure essentially wipes the mesocosm walls and any particles end up suspended in the water column, or if they are sufficiently heavy, they sink into the sediment trap. Either way, these are included in the suspended POM pools (i.e. POC, PON and POP measurements), or in the sinking material collected in the sediment trap which is analysed for the same elements. We will add a clarification to the manuscript on the cleaning procedures.

*Author response: Please see our response above for the changes made to lines 141-144.*

4. The sediment trap setup is interesting. Did you measure organic species other than PIC? How did your routine cleaning impact organic matter resuspension. I assume the mass balance should also include the C and N pools in the sediment trap.

Author response: Yes, the same pools (POC, PON, POP, BSi) were also measured in the sediment trap material collected. These data are reported in a manuscript in this same Special Issue by Süßle et al. (https://egusphere.copernicus.org/preprints/2023/egusphere-2023-2800/).

*Author response: Citation for Suessle et al. has been added to line 144.*

5. I am not fully convinced by the statement in Lines 435-440: "This study contrasts with previous studies where precipitation nuclei started in the water column as suspended particles". It is very possible that the precipitation formed in the water column first and then accumulated and attached to the wall, with cleaning speeding up the process by providing more nuclei. The paper does not have a clear timeline indicating which occurred first. Of course I may misunderstand your point, but either way, please clarify this point.

Author response: We cannot know for sure where the particle originated as we did only look at bulk particle properties from the water column. Reviewer #1 also asked a related question to particle load, so while, unfortunately, no analysis of particle load was made, we will certainly consider this for future studies. Here, the white film that were visibly attached to the mesocosm walls suggest that precipitation at some stage occurred on the walls, but we do agree that this may not be the nucleation site. It was just very obvious.

The experiments reported in the paper by Moras et al. (2022) were in small bottles where the bottle walls could not be wiped clean in the same way as these larger mesocosm units could. The precipitated particles were also clearly visible in the bottles, whereas the same type of observations were not made in this study.

*Author response: We modified lines 448-451 in the revised manuscript as follows:" Cleaning procedures could have then simply removed the precipitated calcium carbonates from the walls and transformed these to particles suspended and sinking in the water column. This apparent primarily benthic origin is in contrast to previous studies in incubation bottles (~ 2L) where the precipitation nuclei were first observed in the water column as suspended particles (Moras et al., 2022)."*

6. Heterotrophy process: It is unclear why PON changes represent heterotrophy. Phytoplankton also contain PON, correct? Did you measure specific heterotrophic species that are sensitive to OAE to get this conclusion?

Author response: Yes, we agree that phytoplankton contain PON, but here we note that the PON was lower at higher OAE, and that inorganic nutrient concentrations, which primarily photoplankton use, were very low. This favour heterotrophic organisms, or mixotrophs that can utilise organic carbon pools. We did not measure specific heterotrophic species to understand their specific sensitivity, however we would refer you to Marin-Samper et al. (https://bg.copernicus.org/articles/21/2859/2024/) in this Special Issue for a study on the overall metabolic balance between autotrophy and heterotophy in this study, as some shifts connected with the plankton community were observed.

*Author response: We do not have any good data to support this however, did add a sentence to lines 486-487 in the revised manuscript to indicate a higher degree of speculation: "This may indicate that the response of heterotrophs to OAE may be relevant for the stability of an alkalinity addition."*

7. The method mentions 13C-DIC, but I did not find any results or discussion about the transfer of carbon throughout the food web. Could you include this information.

Author response: Thank you for picking this up. As we mentioned in response to comments from Reviewer #1, we included this treatment in the methods section for completeness, but the results and discussion thereof are in an accompanying manuscript by Smith-Sanchez et al. (in preparation). We will remove the references to 13C in the manuscript as these are not relevant to this manuscript.

*Author response: We have removed the reference to the $DI^{13}C$ addition from Section 2.2 and in Figure 1.*

Minor comment:

It is unclear what "coastal environments with high surface areas and calcium carbonate structure" means. Please clarify this statement.

Author response: Here we are referring to coastal areas that have relatively shallow water depths and rocks (including carbonate minerals), coastal vegetation, calcifying organisms such as mussels, particle loads from rivers and other structures such as concrete are present in the upper ocean layer, where alkalinity would be likely added. Compared to offshore ocean areas where in the surface layer does not include any rocks or other major substrates, the relative surface area where precipitation could be initiated is much higher. We can add such information to this section of the manuscript during revision.

*Author response: We have added this additional information to lines 501-505 in the revised manuscript to read as follows (new text underlined): "It is also not yet clear if coastal environments with high surface areas for nucleation on shallow submerged rocks or coastal vegetation, presence of calcium carbonate structures and calcifying organisms (e.g. mussels, oysters), and particle loads from riverine inputs may also facilitate precipitation. Compared to offshore ocean areas where the surface layer does not include rocks or other major substrates, the relative surface area in coastal ecosystems where precipitation could be initiated is much higher."*